# A Comparison of Tissue Adhesive Material and Suture as Wound-Closure Techniques following Carpal Tunnel Decompression: A Single-Center Randomized Control Trial

**DOI:** 10.3390/jcm12082864

**Published:** 2023-04-14

**Authors:** Veridijana Sunjic Roguljic, Luka Roguljic, Vedran Kovacic, Ivana Jukic

**Affiliations:** 1Surgery Department, Plastic, Reconstructive and Aesthetic Surgery with Burn Care Division, University Hospital of Split, 21000 Split, Croatia; 2Surgery Department, Orthopaedics and Traumatology Division, University Hospital of Split, 21000 Split, Croatia; 3Internal Medicine Department, Division of Emergency and Intensive Medicine with Clinical Pharmacology and Toxicology, University Hospital of Split, 21000 Split, Croatia; 4School of Medicine, University of Split, 21000 Split, Croatia; 5Internal Medicine Department, Gastroenterology Division, University Hospital of Split, 21000 Split, Croatia

**Keywords:** carpal tunnel, skin adhesive, cyanoacrylate, skin suture

## Abstract

Background: Carpal tunnel syndrome (CTS) is the most common peripheral neuropathy caused by compression of the median nerve in the carpal tunnel. The presented study aimed to evaluate clinical outcomes by comparing two techniques of wound closure following carpal tunnel surgery in subjects randomized to the application of tissue adhesive or sutures. Methods: From April 2022 to December 2022, a single-center randomized prospective trial was conducted at the University Hospital of Split in Croatia. The study participants consisted of 100 patients (70 females) aged 61.56 ± 12.03 years, randomly assigned to suture-based wound closure (*n* = 50) or tissue adhesive-based wound closure (*n* = 50) with two-component skin adhesive Glubran Tiss 2^®^. The outcomes were assessed postoperatively during the follow-up period at intervals of 2, 6, and 12 weeks. A scar assessment was performed using the POSAS (Patient and Observer Scar Assessment Scale) and cosmetic VAS (Visual Analog Scale). The VNRS (Verbal Number Rating Scale) was used to assess pain. Results: There were significant differences between glue-based wound closure and suture-based wound closure at 2-week and 6-week intervals after the surgery on the POSAS and cosmetic-VAS scales (better aesthetic effect with glue-based wound closure technique where noticed), with less postoperative pain at the same intervals. With the 12-week interval, differences in outcomes were insignificant. Conclusions: This trial demonstrated that cyanoacrylate-based adhesion mixtures might be possibly superior in the short term in terms of cosmetic appearance and discomfort compared to conventional skin suturing techniques for the closing of surgical wounds following open CTS decompression, but there was no difference between both procedures in the long term.

## 1. Introduction

Carpal tunnel syndrome (CTS) is the most common peripheral neuropathy caused by compression of the median nerve in the carpal tunnel. From the start, CTS is a potentially debilitating syndrome. In mild to moderate cases, conservative therapy is suggested, while severe cases are treated surgically [1,2]. Patients who have severe carpal tunnel syndrome or whose symptoms have not improved after four to six months of conservative treatment may be considered for surgical decompression [3].

Despite the fact that postoperative complications are uncommon, potential sequelae have a significant impact on patients’ impairment. The most important late complications include persistent symptoms, scars, neurovascular damage, pillar pain, and decreased grip strength [4].

In order to reduce the frequency of late postsurgical complications, various modifications of postsurgical wound closure techniques following carpal tunnel release have been proposed [5]. As postoperative scar tenderness can be induced by inversion of the wound edges, well-performed wound closure will result in reduced postoperative scar tenderness [6]. Sutures, staples, and sticky tapes have been used for many years to seal postoperative wounds.

Tissue adhesives have recently entered clinical practice for these purposes. Due to their strong tensile strength, bacteriostatic characteristics and spontaneous peeling, tissue adhesives such as 2-octylcyanoacrylate are becoming increasingly popular for reinforcing wound closures [7]. However, it has not been proven that there are clear differences between tissue adhesives and classical wound closure techniques in terms of dehiscence, infection, cosmetic appearance, or surgeon and patient satisfaction [8].

Despite many proposed wound-closure techniques following open carpal tunnel decompression surgery, there is a lack of information on the possible role of tissue adhesives in this clinical setting. The presented study aimed to evaluate clinical outcomes by comparing two techniques of wound closure following carpal surgery in subjects randomized to the application of tissue adhesive or sutures. The primary endpoint of the study was aesthetic outcomes, with postoperative pain and the frequency of postprocedural complications as secondary outcomes.

## 2. Materials and Methods

A single-center randomized prospective controlled interventional follow-up study was conducted from 1 April 2022, to 31 December 2022, at the Surgery Department, Plastic, Reconstructive and Aesthetic Surgery with Burn Care Division of the University Hospital of Split in Croatia. The study was designed as a single-blind trial because the outcome evaluators were blinded.

The study protocol was approved by the Ethics Committee at the University Hospital of Split with an ethics code of 500-03/22-01/41; date of approval, 31 March 2022. The study was conducted according to the principles of the Declaration of Helsinki and the Consolidated Standards of Reporting Trials (CONSORT) guidelines. Informed consent was obtained from all subjects involved in the study. This clinical trial was registered with www.clinicaltrials.gov (accessed on 1 March 2023) (NCT05747989).

### 2.1. Study Population

The study population consisted of adult patients (age > 18 years) who had previously been diagnosed with carpal tunnel syndrome and were scheduled for decompression surgery in our department. The diagnosis was based on history, physical examination (weakness of thumb abduction with atrophy of the thenar), and neurological examination with measurement of nerve conduction velocity. The surgery inclusion criteria were the complete failure of conservative treatment for more than 6 months with significant disabilities, such as thenar atrophy, thumb abduction weakness, or severe median nerve conduction impairment estimated by electromyography. The indication for surgery was made by the attending plastic surgeon during the visit, and the patient was scheduled for surgery the following month. Exclusion criteria were threatening hemorrhagic complications (patients with peroral anticoagulation and/or antithrombotic therapy), previous wrist trauma or surgery on the wrist region, another etiology of neuropathy, previous allergic reactions (with lidocaine, cyanoacrylate, formaldehyde, tapes, or adhesives), personal or family history of keloids or hypertrophic scars, and severe general illness with cachexia.

### 2.2. Study Flow

During the study period, 121 patients were recruited, and finally 100 patients were enrolled in the trial after applying exclusion criteria to 19 patients and 2 patients declining to participate. During the 12-week postoperative period, no subjects dropped out or were lost to follow-up. (Figure 1). The size of the sample was calculated considering the anticipated main outcome measure (cosmetic VAS) from previous reports with an alpha error of 0.05 to recognize a significant difference and 90% test power (beta error of 0.1). For this study power, the minimum number of subjects was estimated at 47 in each study group.

The patients were randomly assigned to suture-based wound closure or tissue adhesive-based wound closure. The randomization was generated by a computer as random numbers in a 1:1 ratio between the two interventions. Participants and plastic surgeons were blinded to the intervention until they entered the operating room. Postoperative care and follow-up visits were the same regardless of the intervention group. None of the subjects experienced any side effects from the medication.

### 2.3. Intervention Protocols

All surgical procedures were performed with a tourniquet and local anesthesia using 2% lidocaine in the palmar soft tissues and carpal tunnel. For all subjects, the standard carpal canal decompression procedure began with a 15–18 mm skin incision in the radial half of the palm, followed by carpal ligament transection and cutting [9]. Following primary closure, two different techniques were used depending on the subject’s randomization group:The skin is stitched with transcutaneous nylon sutures (polypropylene-polyethylene monofilament, non-absorbable surgical suture) 4-0. (Optilene^®^ DSMP 19, 3/8 needle, thread size 4/0, B. Braun Surgical, S.A. Carretera de Terrassa, Spain) (Figure 2A).After subcutaneous buried running continuous stitch with 4-0 Coated VicrylTM Plus PS-2, 3/8 (Ethicon Inc., Cincinnati, OH, USA), a two-component skin adhesive, Glubran Tiss 2^®^ (GEM S.r.l., Viareggio, Italy), was applied. Glubran Tiss 2^®^ is composed of NBCA (n-butyl 2 cyanoacrylate) and OCA (2-octyl cyanoacrylate) as a synthetic surgical glue with hemostatic, adhesive, sealing, and bacteriostatic properties [10]. When applied to wet tissue, it immediately polymerizes into a thin, elastic film with a great tensile strength that clings securely to the architecture of the tissue. Polymerized glue is a bioinert material. Each subject received 0.35 mL of Glubran Tiss^®^ on the open wound, and before bandaging, subjects rested for 20 s for a polymerization process (Figure 2B).

Postoperative care consisted of the application of a compressive bandage for 1 day and the introduction of analgesics. Drainage with a narrow plastic tube has been placed if necessary in the first two postoperative days. Regular visits by the attending surgeon and the nurse with wound dressings were done on a daily basis.

### 2.4. Estimation of Outcomes

We assessed the outcomes during the follow-up period at intervals of 2, 6, and 12 weeks postoperatively. During the follow-up, all patients were photographed, completed the VAS and POSAS questionnaires, and completed the VNRS form, while an examiner completed the POSAS questionnaires.

A scar assessment was performed using the Patient and Observer Scar Assessment Scale (POSAS) from both the patient’s and the surgeon’s perspectives. The POSAS is made up of two scales: the patient scale and the observer scale; each of the six components is scored numerically on a scale of 1 to 10. The component scores are then added together; the worst scar would receive a score of 60, while the best scar would receive a score of 6 [11]. Subjects filled out a standardized scar assessment form (POSAS), and blinded photos were collected so that an independent observer (a surgeon) could evaluate the scar as a POSAS score. Independent observers were blinded and did not participate in the intervention or follow-up examinations.

A Verbal Number Rating Scale (VNRS) was used to assess the degree of pain in the hand before and the day after surgery, as well as at 2, 6, and 12-week intervals during the follow-up period. The VNRS is a verbal self-report instrument with a 0–10 numeric rating scale, where 0 represents no pain and 10 represents the most severe pain possible [12]. Additionally, a cosmetic VAS (Visual Analog Scale) assessment form was filled out by the patient at the 2, 6, and 12-week intervals. The cosmetic-VAS is a 0–100 scale with “worst scar” written at the left end (0) and “best scar” written at the right end (100) [13].

Gender, weight, height, previous illnesses, length of postoperative wound, nerve decompression time, stitching time, and bandage time were recorded for each subject.

### 2.5. Statistical Analysis

Descriptive statistics calculations and data were expressed as the arithmetic mean ± standard deviation if normally distributed, or as the median (interquartile range) if not normally distributed. The Kolmogorov–Smirnov test was used for the estimation of the normality of quantitative variables’ distributions. Qualitative data between groups were compared with Fisher’s exact tests. Quantitative data were compared using an unpaired Student’s *t*-test. The Mann–Whitney test was employed to analyze and compare quantitative variables with non-normal distribution. Correlations between quantitative data were calculated as the significance of the Pearson’s correlation coefficient for normally distributed variables or of the Spearman’s rho coefficient for variables with a non-normal distribution. Statistical analysis was performed with SPSS software for Windows (IBM SPSS Statistics for Windows, version 26.0, Armonk, NY, USA). *p* values < 0.05 were considered significant.

## 3. Results

The study participants consisted of 100 patients (30 males and 70 females) randomly assigned in a 1:1 ratio for wound closure with glue (*n* = 50) and wound closure with sutures (*n* = 50). The age of the entire cohort was 61.56 ± 12.03 years. The right-side surgery was performed on 57 subjects, while the left-side surgery was performed on 43. The mean surgery decompression time for CTS resolution was 8.71 ± 0.57 min. Subjects’ clinical presentation with surgical times and outcomes, along with gender differences, is demonstrated in Table 1. The differences between patients whose wounds were closed with glue and those whose wounds were closed with sutures are presented in Table 2.

Postoperative complications after 15 days in the entire cohort were observed in 12 participants: redness (5 cases), dehiscence (2), hematoma (2), infection (1), allergic dermatitis (1), and secretion (1 case). There were no statistically significant differences in the number of complications between the glue-based wound closure and the suture-based wound closure groups of the patients (Chi square < 0.001, *p* = 0.620). After 12 weeks of surgery, we observed complications in 8 subjects, granuloma in 5 cases, secretion in 2, and infection in 1. There were no statistically significant differences in the number of complications between the glue-based wound closure and suture-based wound closure groups of the patients (Chi square = 0.54, *p* = 0.375). We found a weak but significant correlation between surgery time and wound length (r = 0.206, *p* = 0.020). At 6 weeks, the VNRS score was weakly correlated with the bandaging time (rho = −0.347, *p* < 0.001). BMI was weakly negatively correlated with the observer’s POSAS score at 6 weeks (rho = −0.270, *p* = 0.03).

A correlation analysis between BMI and outcome scores in the subgroup of glue wound closure subjects is demonstrated in Table 3. Moderate significant correlations were found between the BMI and POSAS scores of the patient and the observer in the 6th week. Figure 3 shows a regression graph with regression equation between the BMI and the POSAS score of the patient in the 6th week in the subgroup of glue wound closure subjects. Figure 4 demonstrated the time changes in the POSAS scores evaluated by the patient during the postoperative follow-up period.

## 4. Discussion

The results of this trial demonstrated that the tissue adhesion technique for wound closure is significantly better after 2 and 6 weeks in terms of aesthetic effects than the classic stitch technique in surgically treated patients with CTS. Furthermore, patients subjected to the tissue adhesion technique reported less postoperative pain in the 2nd and 6th weeks after the surgical procedure. Despite the fact that the mentioned differences in postoperative pain were statistically significant, the objectively measured differences are very small, and the clinical significance of such differences in the experience of postoperative pain has objectively little clinical significance. Additionally, although there are statistically significant differences between the two wound closure techniques in the 2nd and 6th weeks, these differences are still clinically insubstantial to have a major influence on the surgeon’s decision as to which technique to use. In addition, after 12 weeks, the mentioned difference in aesthetic effects and postoperative pain is lost, which further diminishes the practical importance of choosing a wound closure technique in real-world clinical practice.

Surgical treatment of CTS is reserved for the subgroup of patients with an insufficient response to conservative therapy or for severe, debilitated cases of CTS with significant functional deficits. Although surgical decompression of the carpal tunnel is a relatively safe treatment, late postoperative problems, such as prolonged discomfort or functional impairments, are nevertheless considerable. Boya et al. [14]. reported scar tenderness in 7% and pillar pain in 18% of patients subjected to surgical treatment of CTS.

Scar formation after open carpal tunnel release is one of the most common causes of discomfort and functional disability, as this procedure causes deep skin and subcutaneous injuries that can result in hypertrophic scars and keloids [15]. In addition, abnormal scars can decrease the quality of life and deteriorate the social and physical status of the patient [16]. It is not easy to assess the aesthetic effects of surgery. Many techniques have been used in the past to evaluate the aesthetic effects of surgical procedures. The methods and techniques of aesthetic effect evaluation are prone to the subjective evaluations of the observer. The greatest advance in the evaluation of the aesthetic effects of surgery was made in the aesthetic surgery of the face and the neck [17,18].

The aforementioned surgical complications triggered a considerable effort to find a better surgical technique to preserve functional capacity and diminish scar development after CTS surgery. Suwannaphisit et al. [19] demonstrated that the Donati suture resulted in higher POSAS scores compared to the subcuticular running sutures, despite the fact that both techniques generally resulted in low POSAS scores and good scar formation. In this study, the average POSAS score after 2 weeks was lower for the running subcuticular suture (15.3 ± 4.8) compared with the Donati suture (17 ± 4.6), but observer scores were not significantly different (15.6 ± 5.8 vs. 16.7 ± 5.2) after 2, 6, or 12 weeks. These POSAS scores are similar to our results after 2 weeks (median 16 in glue-based group), but in contrast to the cited study, the difference in POSAS score was visible in our subjects even after 6 weeks in both assessment methods. In study by Suwannaphisit et al., the authors could not find differences between the Donati suture and the running subcuticular suture in VNRS pain scores at 2, 6, and 12 postsurgical weeks. The authors found that both suture methods are appropriate for wound closure following an open carpal tunnel surgery. In contrast, we found clear differences in the VNRS score in the 2nd and 6th postoperative weeks, demonstrating the favorability of glue-based tissue adhesion.

A study in CTS patients, although aiming at conserving superficial nerve branches at the incision site during open carpal tunnel decompression, did not diminish the incidence or intensity of postoperative discomfort [20]. At 6 weeks, 3 months, and 6 months, the authors found no proof that the two treatments differed in terms of scar pain. On the contrary, in our cohort, a significant difference in pain assessment was found at 2 and 6 weeks post surgery. In addition to the aforementioned studies, numerous additional efforts were made to advance surgical procedures in CTS patients; however, the findings were inconclusive and failed to demonstrate improvement during postprocedural follow-up [21].

In addition to comparisons of suture or surgery techniques, some studies examined types of suture materials and their impact on postoperative complications after carpal tunnel decompression. There is a lack of definitive conclusions in the comparison of absorbable sutures with non-absorbable sutures after carpal tunnel decompression. In a meta-analysis of five trials, Wade et al. [22] concluded that it was unclear if using absorbable or non-absorbable sutures for skin closure after carpal tunnel decompression caused any differences in discomfort or wound inflammation. In most of the included studies, the visual analogue scale (VAS) was used for pain assessment. In one of those studies [23], the authors compared the aesthetic outcome of scars after closure of open carpal tunnel with either absorbable 4-0 Vicryl Rapide or non-absorbable 4-0 Novafil at 6 weeks using a POSAS score and did not find differences between the two techniques. Contrary to the results of those studies, we showed that after 2 and 6 weeks, clear differences between the two surgical wound closure techniques in relation to pain and aesthetic effect, although the possible practical significance of these differences might be clinically negligible.

In recent years, cyanoacrylate-based skin glue has become more popular in various surgical procedures for ensuring and stabilizing wound closure [24,25]. One of the recognized advantages of these glues is their ability to decrease the risk of surgical site infection by physically isolating the surgical wound [26,27].

Despite its popularity, there is a scarcity of conclusive evidence favoring skin adhesives over sutures. A Cochrane systematic analysis found no difference in aesthetic outcomes between tissue adhesives and conventional wound closure or between different tissue adhesives [28]. Furthermore, a randomized study using a topical skin adhesive (2-octyl-cyanoacrylate) for wound closure in forefoot surgery found that skin adhesive use was related to higher inflammation and regions of wound separation than nylon sutures [29]. In a recent Cochrane systematic analysis of 33 studies involving 2793 participants, sutures were found to be superior to tissue adhesives in preventing wound dehiscence, and no differences were found in wound infection [8]. The same systematic analysis in a subanalysis of trials comparing the use of tissue adhesives with sutures found no evidence of a difference in the participants’ and surgeons’ assessments of cosmetic appearance measured by the cosmetic VAS score. In contrast to the aforementioned trials, a clear difference between the aesthetic postoperative effects measured as VAS scores of glue-based and suture-based wound closure was proven in our subjects.

Although wound closure using skin adhesives has been studied in numerous surgical procedures, the evaluation of skin adhesives following carpal tunnel decompression is still pending. A prospective randomized controlled trial was recently carried out to compare adhesive tape and tissue adhesive applied after primary closure to different halves of the same surgical incision [30]. Most of the incisions (50 in total) were for carpal tunnel decompression (14 subjects) and thumb carpometacarpal arthroplasty (14 subjects). Wounds were initially closed with 4-0 absorbable suture, then the proximal and distal parts of the wounds were finally closured with tapes (Steri Strips) or 2-octylcyanoacrylate (Dermabond). The authors evaluated the scars at approximately 3 months and concluded that the adhesive strips provided a modest but significant improvement in cosmetic outcomes compared to a tissue adhesive (POSAS score) observed by a surgeon. Patient observed better cosmetic outcomes with Steri-Strips than with Dermabond, although differences were not statistically significant.

Sinha et al. [31] conducted a prospective, randomized study with the aim of comparing the outcome of hand surgery wounds repaired with a tissue adhesive (n-butyl 2-cyanoacrylate tissue adhesive) or with standard wound closure techniques (4-0 monofilament suture). The 50 participants had hand surgery; 22 had CTS surgery. The authors could not find a significant difference in the cosmetic outcome assessment in the cohort of participants assessed at 2 and 6 weeks post surgery (mean cosmetic VAS score in the tissue adhesive group was 81 vs. 87 in the suture group).

In contrast to the previously mentioned studies, our randomized controlled trial found that the application of skin adhesion after carpal tunnel decompression brought about a better aesthetic effect and improved patient satisfaction compared to a control group that used sutures as a wound closure procedure. These effects are maintained throughout the second and sixth weeks, and the difference becomes insignificant in the twelfth week. Additionally, the cosmetic VAS score was significantly higher in both our groups than in the study by Sinha et al. The time required for wound closure was shorter in the skin adhesion application group, highlighting the ease of use of commercially available skin glue preparation. However, the results of our study are unlikely to have a significant impact on the surgeon’s clinical decision about which technique to use in wound closure after CTS surgery. Interestingly, in the skin adhesion group, the results of our study have shown a moderate significant negative correlation between BMI and the good aesthetic effects estimated by the patient and the observer. We hypothesized that lower BMI values were associated with malnutrition, which might impact suboptimal aesthetic surgical outcomes.

However, the presented study had certain limitations. Firstly, the follow-up duration was insufficient to determine long-term problems and monitor long-lasting scar development following carpal tunnel release. Secondly, participants and data were restricted to a single center. Third, significant limitation of this study is that the possible effects of the compared wound closure technique on biomechanical and functional postoperative complications were not evaluated.

## 5. Conclusions

This randomized controlled trial found that wound closure following open CTS using cyanoacrylate-based adhesion material had a mild advantage over sutures in terms of aesthetic outcomes, discomfort, and patient compliance at 2 and 6 weeks postoperatively. However, the cosmetic results leveled off at 12 weeks postoperatively with no significant differences. As a conclusion, this trial demonstrated that cyanoacrylate-based adhesion mixtures might be possibly superior in the short term in terms of cosmetic appearance compared to conventional skin suturing techniques for the closing of surgical wounds following open CTS decompression. Larger multicentric studies with additional clinical and functional outcomes are needed to clarify the aforementioned conclusions.

## Figures and Tables

**Figure 1 jcm-12-02864-f001:**
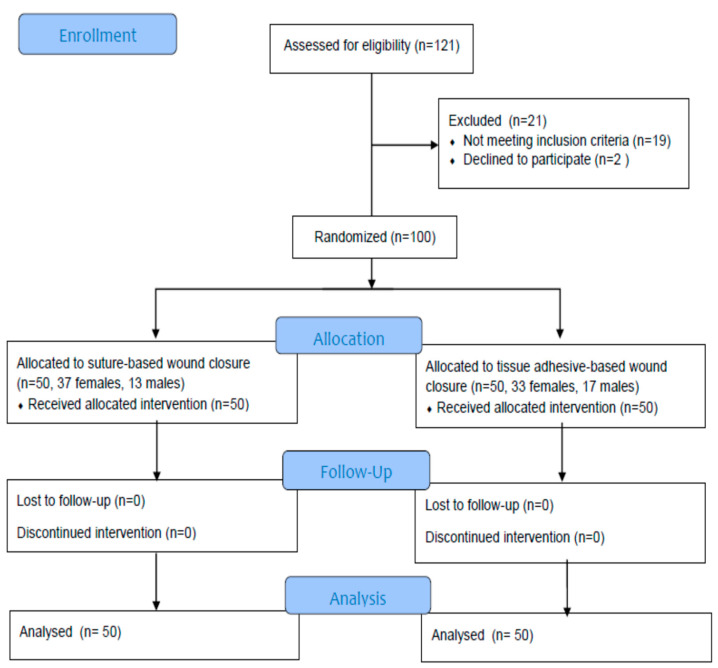
CONSORT flow diagram of the study.

**Figure 2 jcm-12-02864-f002:**
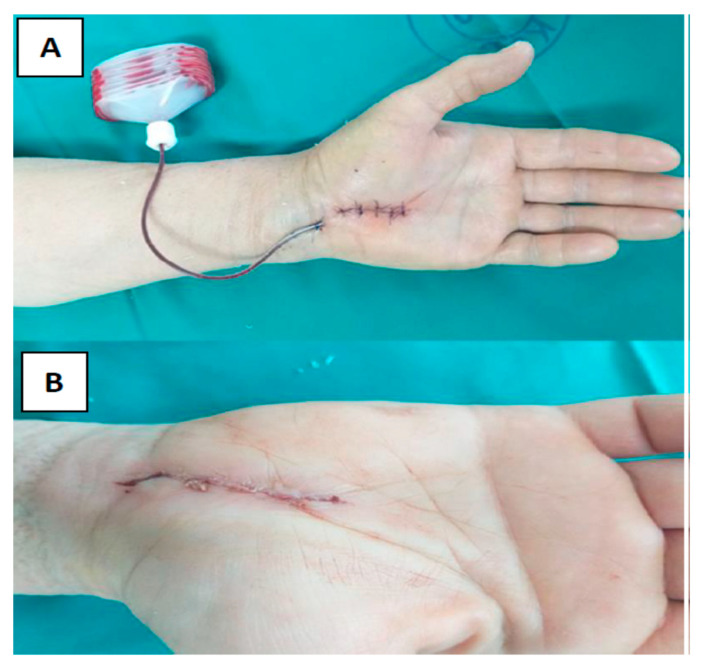
Photographs of a patient demonstrating suture wound closure (**A**) and a patient demonstrating wound closure with Glubran Tiss^®^ (**B**).

**Figure 3 jcm-12-02864-f003:**
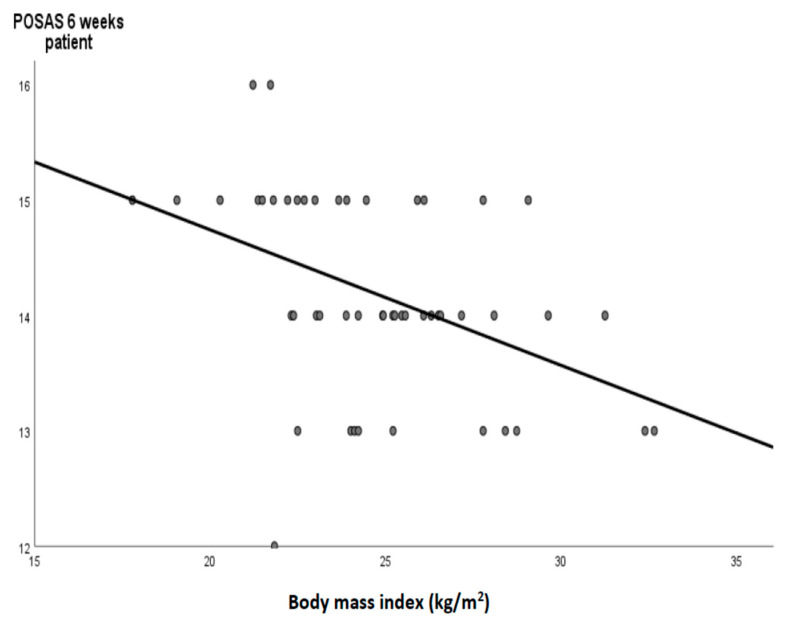
A regression graph (black line) and circles as a cases with the regression equation demonstrated a moderate correlation between the body mass index and POSAS score of the patient in the 6th week in the subgroup of glue wound closure subjects (Y= −0.118X + 17.09).

**Figure 4 jcm-12-02864-f004:**
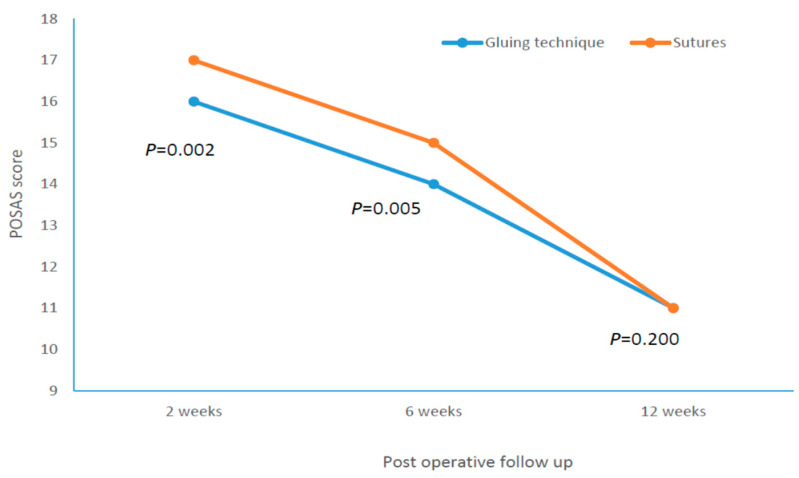
POSAS scores evaluated by the patient during the follow-up period after surgery.

**Table 1 jcm-12-02864-t001:** Subjects’ clinical presentations with surgical times and outcomes. Data are shown for the entire patient population as well as for the two subgroups; sex differences are calculated (Student’s *t*-test for independent samples or Mann–Whitney test for non-parametric data), and *p*-values for significant differences are bolded. Data are presented as the arithmetic mean ± standard deviation, or median (interquartile range) for non-parametric variables.

	All (*n* = 100)		Males (*n* = 30)		Females (*n* = 70)		*p*
Age (years)	61.56 ± 12.03		66.30 ± 10.88		59.53 ± 12.00		** 0.005 **
BMI (kg/m^2^)	24.92 ± 2.74		26.44 ± 2.12		24.26 ± 2.73		** <0.001 **
Time_stitching (min)	3.25 ± 0.31		3.29 ± 0.27		3.23 ± 0.32		0.198
Time_bandage (min)	12.87 ± 1.01		12.70 ± 0.99		12.94 ± 1.02		0.131
Sugical_decompression (min)	8.71 ± 0.57		8.60 ± 0.53		8.75 ± 0.58		0.115
cosmetic-VAS 2 weeks	95.00	(90.00–95.00)	90.00	(90.00–95.00)	95.00	(90.00–95.00)	0.270
cosmetic-VAS 6 weeks	100.00	(95.00–100.00)	100.00	(95.00–100.00)	100.00	(95.00–100.00)	0.159
cosmetic-VAS 12 weeks	100.00	(100.00–100.00)	100.00	(100.00–100.00)	100.00	(100.00–100.00)	0.092
POSAS 2 weeks pt.	17.00	(16.00–17.00)	17.00	(16.00–17.00)	17.00	(16.00–17.25)	0.117
POSAS 2 weeks observer	17.00	(16.00–17.00)	16.00	(16.00–17.00)	17.00	(16.00–17.00)	**0.006**
POSAS 6 weeks pt.	15.00	(14.00–15.00)	14.00	(13.00–15.00)	15.00	(14.00–15.00)	0.143
POSAS 6 weeks observer	14.00	(13.00–15.00)	14.00	(13.00–14.00)	14.00	(13.75–15.00)	0.060
POSAS 12 weeks pt.	11.00	(10.00–12.00)	11.00	(11.00–12.00)	11.00	(10.00–12.00)	0.145
POSAS 12 weeks observer	11.00	(10.00–12.00)	11.00	(10.00–12.00)	11.00	(10.00–12.00)	0.180
VNRS prior surgery	5.00	(4.00–6.00)	5.00	(4.00–6.00)	5.00	(4.00–5.25)	0.403
VNRS on surgery day	5.00	(4.00–6.00)	4.50	(4.00–6.00)	5.00	(4.00–6.00)	0.084
VNRS 2 weeks post surgery	3.00	(3.00–4.00)	3.00	(2.00–4.00)	3.00	(3.00–4.00)	**0.023**
VNRS 6 weeks post surgery	2.00	(1.00–2.00)	2.00	(1.00–2.00)	2.00	(1.75–2.00)	0.087
VNRS 12 weeks post surgery	0.00	(0.00–0.00)	0.00	(0.00–0.00)	0.00	(0.00–0.00)	0.374
wound length (mm)	17.00	(17.00–18.00)	17.00	(17.00–18.00)	18.00	(17.00–18.00)	**0.045**

Legend: BMI: Body Mass Index, pt.: patient, VAS: Visual Analogue Scale, POSAS: Patient and Observer Scar Assessment Scale, VNRS: Verbal Numerical Rating Scale.

**Table 2 jcm-12-02864-t002:** Differences between glue-based wound closure and suture-based wound closure patients (Student’s *t*-test for independent samples or Mann–Whitney test for non-parametric data), and *p*-values for significant differences are bolded. Data are presented as the arithmetic mean ± standard deviation, or median (interquartile range) for non-parametric data.

	Glue-Based Technique (*n* = 50)		Suture-Based Technique (*n* = 50)		*p*
Age (years)	63.02 ± 12.97		60.10 ± 10.95		0.113
BMI (kg/m^2^)	24.79 ± 3.17		25.04 ± 2.25		0.325
Time_stitching (min)	3.19 ± 0.27		3.31 ± 0.33		**0.021**
Time_bandage (min)	12.93 ± 1.00		12.81 ± 1.03		0.291
Sugical_decompression (min)	8.74 ± 0.57		8.67 ± 0.57		0.264
cosmetic-VAS 2 weeks	95.00	(90.00–95.00)	90.00	(90.00–95.00)	**0.014 **
cosmetic-VAS 6 weeks	100.00	(100.00–100.00)	100.00	(95.00–100.00)	**0.003 **
cosmetic-VAS 12 weeks	100.00	(100.00–100.00)	100.00	(100.00–100.00)	0.153
POSAS 2 weeks pt.	16.00	(16.00–17.00)	17.00	(17.00–18.00)	**0.002 **
POSAS 2 weeks observer	16.00	(16.00–17.00)	17.00	(16.00–18.00)	**<0.001 **
POSAS 6 weeks pt.	14.00	(14.00–15.00)	15.00	(14.00–15.00)	**0.005 **
POSAS 6 weeks observer	14.00	(13.00–14.00)	14.00	(13.75–15.00)	**0.038 **
POSAS 12 weeks pt.	11.00	(10.00–12.00)	11.00	(10.00–12.00)	0.200
POSAS 12 weeks observer	11.00	(10.00–11.25)	11.00	(10.00–12.00)	0.064
VNRS prior surgery	5.00	(4.00–6.00)	5.00	(4.00–5.25)	0.387
VNRS on surgery day	5.00	(4.00–6.00)	5.00	(4.00–6.00)	0.134
VNRS 2 weeks post surgery	3.00	(3.00–4.00)	3.00	(3.00–4.00)	**0.027 **
VNRS 6 weeks post surgery	2.00	(1.00–2.00)	2.00	(2.00–3.00)	**0.001 **
VNRS 12 weeks post surgery	0.00	(0.00–0.00)	0.00	(0.00–0.00)	0.232
wound length (mm)	17.00	(17.00–18.00)	17.00	(17.00–18.00)	0.355

Legend: BMI: Body Mass Index, pt.: patient, VAS: Visual Analogue Scale, POSAS: Patient and Observer Scar Assessment Scale, VNRS: Verbal Numerical Rating Scale.

**Table 3 jcm-12-02864-t003:** Correlation analysis between body mass index (BMI) and outcome scores in a subgroup of subjects with glue wound closure (*n* = 50); Spearman’s correlation, one-tailed, significant correlations are bolded.

	Spearman’s Rho	*p*
wound length	−0.088	0.273
Time_stitching	0.085	0.279
Time_bandage	0.016	0.456
Sugical_decompression	**0.239 **	**0.048 **
cosmetic-VAS 2 weeks	−0.183	0.101
cosmetic-VAS 6 weeks	**−0.290 **	**0.021 **
cosmetic-VAS 12 weeks	−0.064	0.329
POSAS 2 weeks pt.	**−0.252 **	**0.039 **
POSAS 2 weeks observer	**−0.294 **	**0.019 **
POSAS 6 weeks pt.	**−0.460 **	**<0.001 **
POSAS 6 weeks observer	**−0.407 **	**0.002 **
POSAS 12 weeks pt.	−0.005	0.487
POSAS 12 weeks observer	0.104	0.237
VNRS prior surgery	0.062	0.335
VNRS on surgery day	−0.230	0.054
VNRS 2 weeks post surgery	**−0.355 **	**0.006 **
VNRS 6 weeks post surgery	−0.208	0.074
VNRS 12 weeks post surgery	0.219	0.063

Legend: pt.: patient, VAS: Visual Analogue Scale, POSAS: Patient and Observer Scar Assessment Scale, VNRS: Verbal Numerical Rating Scale.

## Data Availability

The data presented in this study are available on request from the corresponding author.

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
