# Peer review of "A Comparison of Tissue Adhesive Material and Suture as Wound-Closure Techniques following Carpal Tunnel Decompression: A Single-Center Randomized Control Trial"

_jcm, 2023, doi:10.3390/jcm12082864_

Round 1
Reviewer 1 Report
Dear author.
Thank you for the opportunity to review this article on wound closure in carpal tunnel syndrome. I have read it with great interest.
However, at present, the manuscript includes several issues that need to be resolved, and I would like to ask the author to resolve them.
1. The author described that the difference in POSAS is 1 or 2 points in Table 2, which introduced the conclusion that the tissue adhesive-based wound closure was superior in terms of cosmetic appearance. Is the difference CLINICALLY significant enough to conclude that the procedure is clearly superior in terms of appearance? Even if there is a statistically significant difference, 1 or 2 points of difference in POSAS seems not to be significant enough to change the decision-making of the surgeon.
Statistical significant differences are likely to occur as the increase of sample sizes. Conclusions should not be based solely on statistically significant differences. The same can be said for differences in VNAS. In my opinion, the study seems to show no CLINICALLY significant differences between the two wound closure procedures.
For example, in a population taking a newly developed antihypertensive drug in a clinical trial, blood pressure fell by 2 mmHg in the group of patients who took the drug compared to the control group, and there was a statistical difference between the two groups. Would you consider 2 mmHg of blood pressure, in this case, to be a clinically significant difference enough to select the new drug?
2. The authors described that tissue adhesive-based wound closure following the subcutaneous sutures was better in postsurgical pain than the typical suture-based wound closure. In the typical suture-based wound closure, sutures are usually removed 1-2 weeks after surgery. On the other hand, subcutaneous absorbable sutures are not removed. Threads of subcutaneous sutures (Biclyl) are reported to take up to six months to resolve and absorb. Inflammation of the subcutaneous tissue can occur more often in the cases of sutured with the subcutaneous absorbable thread after surgery than in suture-based wound closure. It is difficult to imagine that postoperative pain would be lower with subcuticular sutures compared to monofilament skin sutures, but how does the author explain that?
3. Some hypothesize that postoperative pain after carpal tunnel release is not only caused by the wound scar, but also by dynamic changes in the carpal bone due to the incision of the transverse carpal ligament. If it is only caused by the wound scar, then one-portal endoscopic surgery (only 7-10 mm of incision proximal to the wrist) should cause little pain. Actually, however, many studies have shown that there is no difference in postoperative pain depending on the closure. If this is the case, why did the suture method make a difference in this study? It would be more reasonable to interpret the results as a statistical difference but not a clinical difference.
4. The results for cosmetic VAS are not stated at all in the tables and figures. The abstract and text state that there was a difference, but if you have done the analysis, the results need to be clearly stated.
5. Figure 2 is a very clear surgical operation photograph to show the characteristics of the two groups. The upper and lower rows are described in the figure caption, but it becomes easier to understand for many readers if they are marked with A and B, and linked to the links in the text (lines 123 and 132).
6. In the abstract and conclusion, the author described that cyanoacrylate-based tissue adhesion mixtures are safe easy to use, and efficient for CTS open surgery. This study, however, did not prove that "cyanoacrylate-based adhesion mixtures" are safe and easy to use. The authors did not conduct research on safety verification or the feeling of use at all. They only show that closure with cyanoacrylate-based adhesion mixtures after subcutaneous suturing may be possibly superior in the short term in terms of cosmetic appearance compared to conventional skin suturing techniques and that there is no difference between both procedures in the long term. Both descriptions should be corrected.
Author Response
Response to Reviewer’s 1 Comments
- The author described that the difference in POSAS is 1 or 2 points in Table 2, which introduced the conclusion that the tissue adhesive-based wound closure was superior in terms of cosmetic appearance. Is the difference CLINICALLY significant enough to conclude that the procedure is clearly superior in terms of appearance? Even if there is a statistically significant difference, 1 or 2 points of difference in POSAS seems not to be significant enough to change the decision-making of the surgeon.
Statistical significant differences are likely to occur as the increase of sample sizes. Conclusions should not be based solely on statistically significant differences. The same can be said for differences in VNAS. In my opinion, the study seems to show no CLINICALLY significant differences between the two wound closure procedures.
For example, in a population taking a newly developed antihypertensive drug in a clinical trial, blood pressure fell by 2 mmHg in the group of patients who took the drug compared to the control group, and there was a statistical difference between the two groups. Would you consider 2 mmHg of blood pressure, in this case, to be a clinically significant difference enough to select the new drug?
Response 1: We completely agree with the Reviewer’s suggestion and change sections Discussion and Conclusion accordingly. Additional paragraphs and sentences are added. We explain clinically inessential differences between the two techniques in weeks 2. and 6. , and emphasises that results of the study should be interpreted to mean that cyanoacrylate-based adhesion mixtures are not inferior to sutures.
- The authors described that tissue adhesive-based wound closure following the subcutaneous sutures was better in postsurgical pain than the typical suture-based wound closure. In the typical suture-based wound closure, sutures are usually removed 1-2 weeks after surgery. On the other hand, subcutaneous absorbable sutures are not removed. Threads of subcutaneous sutures (Biclyl) are reported to take up to six months to resolve and absorb. Inflammation of the subcutaneous tissue can occur more often in the cases of sutured with the subcutaneous absorbable thread after surgery than in suture-based wound closure. It is difficult to imagine that postoperative pain would be lower with subcuticular sutures compared to monofilament skin sutures, but how does the author explain that?
Response 2: We completely agree with the Reviewer’s recommendation, and in the Intervention protocol subsection, we explained that subcutaneous stiches were buried running continuous stitches.
We additionally want to explain the issue: Monofilament skin sutures include a larger amount of tissue on both sides of the wound, wider at the base of the suture (the dermis) than at the surface of the skin (the epidermis), which causes compression at the base of the suture and pressure of the tissue outwards, resulting in a slight twisting of the edges of the wound. Sensory receptors (nociceptors) are located in the dermis and extend into the epidermis and are compressed by monofilament skin sutures.
On the other hand, buried running dermal-subcutaneous sutures pass only subcutaneously and thus relieve the tension of the wound. Namely, because there are no sensitive receptors (nociceptors) in the subcutaneous tissue, subcutaneous sutures have a greater ability to support the edges of the wound and thus enable a better distribution of tensile forces.
- Some hypothesize that postoperative pain after carpal tunnel release is not only caused by the wound scar, but also by dynamic changes in the carpal bone due to the incision of the transverse carpal ligament. If it is only caused by the wound scar, then one-portal endoscopic surgery (only 7-10 mm of incision proximal to the wrist) should cause little pain. Actually, however, many studies have shown that there is no difference in postoperative pain depending on the closure. If this is the case, why did the suture method make a difference in this study? It would be more reasonable to interpret the results as a statistical difference but not a clinical difference.
Response 3: We fully support the reviewer's suggestion and the accordingly revised section Discussion. We add a sentence related to the little clinical difference, despite the statistical difference.
- The results for cosmetic VAS are not stated at all in the tables and figures. The abstract and text state that there was a difference, but if you have done the analysis, the results need to be clearly stated.
Response 4: We completely agree with the Reviewer’s suggestion and clarified and stated cosmetic-VAS (Visual Analog Scale) scores during follow up with calculated differences in all of presented Tables.
- Figure 2 is a very clear surgical operation photograph to show the characteristics of the two groups. The upper and lower rows are described in the figure caption, but it becomes easier to understand for many readers if they are marked with A and B, and linked to the links in the text (lines 123 and 132).
Response 5: As the Reviewer suggested, we marked parts of Figure 2 as A and B. Additionally, we linked the new Figure markers with associated text in the subsection Intervention protocols.
- In the abstract and conclusion, the author described that cyanoacrylate-based tissue adhesion mixtures are safe easy to use, and efficient for CTS open surgery. This study, however, did not prove that "cyanoacrylate-based adhesion mixtures" are safe and easy to use. The authors did not conduct research on safety verification or the feeling of use at all. They only show that closure with cyanoacrylate-based adhesion mixtures after subcutaneous suturing may be possibly superior in the short term in terms of cosmetic appearance compared to conventional skin suturing techniques and that there is no difference between both procedures in the long term. Both descriptions should be corrected.
Response 6: We fully support the reviewer's suggestion and the accordingly revised sections Abstract and Conclusion exactly as the reviewer recommended.
Reviewer 2 Report
The study aimed to compare the effects of two wound closure techniques (tissue adhesive material and suture) on clinical outcomes following open carpal tunnel decompression surgery in a single-center randomized single-blind control trial. The cyanoacrylate-based adhesion material had an advantage over sutures in terms of aesthetic outcomes and patient discomfort. The novelty of the study is the application of an existing tissue adhesive-based wound closure technique to carpal tunnel decompression surgery, for achieving better postoperative outcomes. The proposed method of wound closure limits the shortcomings of open carpal tunnel surgery and has immense potential for providing better patient care. However, there are other surgical techniques available (eg, endoscopic CTR, minimally invasive CTR, ultra-minimal thread CTR, etc.) that minimize scar formation and address the limitations of open carpal tunnel decompression surgery. Hence, the clinical applicability and significance of the proposed method are limited in the field of carpal tunnel decompression.
The study is very well written, and the experimental design is rigorous. However, the manuscript can be further improved based on the following suggestions:
· Correct typo in the abbreviation of carpal tunnel syndrome in Lines 41, 246, and 247.
· The reference listed in Line 43 is for conservative therapy. Please also include a reference for surgical therapy.
· Line 80 – Interchange the words “was” & “trial”
· Lines 99 & 100 – Sentence can be restructured to improve clarity
· Figure 1: Please provide the counts of Males and Females within each wound closure technique after randomized allocation
· Correlation analysis:
o Please mention the strength of each correlation (strong, moderate, or weak) in either results or in Table 3 so that the readers can interpret the analysis accordingly.
o Line 221 mentions that there is a linear correlation between the BMI and POSAS 6-week patient scores, but Figure 3 doesn’t demonstrate linearity in the data. Based on Table 3, the strength of the correlation is moderate at best, with most of the correlation being weak. Keeping that in mind even though some correlations in Table 3 are statistically significant, they do not hold much clinical significance. The correlation analysis should be cautiously interpreted, especially with a weak correlation.
· Lines 259-261 can be restructured for better clarity
· Discussion:
o The discussion section provides a good literature review of previously used wound closure techniques and their performance. However, it seems like most of the section is just an expansion of the introduction, with only one paragraph talking about the results of the current study. The section must be improved significantly by providing an interpretation of results, comparing POSAS, VAS, and VNRS results with previous studies, discussing the performance of the proposed technique (i.e. postoperative complications observed), and expanding on clinical significance (i.e. the proposed technique provides early pain relief compared to conventional technique).
o The authors should add a limitation that the effects of the proposed technique on the biomechanical and functional postoperative complications were not evaluated.
Author Response
Response to Reviewer’s 2 Comments
- Correct typo in the abbreviation of carpal tunnel syndrome in Lines 41, 246, and 247.
Response 1. As the Reviewer suggested, we corrected abbreviation of carpal tunnel syndrome (CTS) across the text.
- The reference listed in Line 43 is for conservative therapy. Please also include a reference for surgical therapy.
Response 2. We completely agree with the Reviewer’s suggestion and added a reference regarding surgical therapy of carpal tunnel syndrome.
- Line 80 – Interchange the words “was” & “trial”
Response 3. According Reviewer’s suggestion, we corrected word order in line 80 (“was” & “trial”).
- Lines 99 & 100 – Sentence can be restructured to improve clarity
Response 4. According Reviewer’s recommendation, we amended and restructured the mentioned sentence on the beginning of the Study flow subsection.
- Figure 1: Please provide the counts of Males and Females within each wound closure technique after randomized allocation
Response 5. According Reviewer’s recommendation, we changed Figure 1 and provide counts of males and females in groups.
- Please mention the strength of each correlation (strong, moderate, or weak) in either results or in Table 3 so that the readers can interpret the analysis accordingly.
Response 6. According Reviewer’s recommendation we mentioned and explained the strength of significant correlations (weak and moderate).
- Line 221 mentions that there is a linear correlation between the BMI and POSAS 6-week patient scores, but Figure 3 doesn’t demonstrate linearity in the data. Based on Table 3, the strength of the correlation is moderate at best, with most of the correlation being weak. Keeping that in mind even though some correlations in Table 3 are statistically significant, they do not hold much clinical significance. The correlation analysis should be cautiously interpreted, especially with a weak correlation.
Response 7. According to the reviewer's suggestion, we restructured the paragraph in the Results section regarding the correlation between the BMI and POSAS 6-week patient scores and the described regression graph, with additional explanation in the Discussion section.
- Lines 259-261 can be restructured for better clarity
Response 8. According to the reviewer's suggestion, we restructured the mentioned sentence between lines 259-261.
- The discussion section provides a good literature review of previously used wound closure techniques and their performance. However, it seems like most of the section is just an expansion of the introduction, with only one paragraph talking about the results of the current study. The section must be improved significantly by providing an interpretation of results, comparing POSAS, VAS, and VNRS results with previous studies, discussing the performance of the proposed technique (i.e. postoperative complications observed), and expanding on clinical significance (i.e. the proposed technique provides early pain relief compared to conventional technique).
Response 9. We completely agree with the Reviewer’s suggestion and substantially change sections Discussion. According to ecommendations, we significantly revised this section, discussed the significance of the previous studies, and compared with interpretations of our results (aesthetic and pain outcomes and scores). In order to expand this section according reviewer’s suggestions, we have added several new paragraphs and several new references. Also, we added explanations regarding the clinical significance of our results.
- The authors should add a limitation that the effects of the proposed technique on the biomechanical and functional postoperative complications were not evaluated.
Response 10. According Reviewer’s recommendation, we changed the end of the Discussion section and added a sentence about limitations regarding possible biomechanical and functional postoperative complications compared with different surgical techniques.
Round 2
Reviewer 1 Report
All my comments seem to be appropriately addressed and the manuscript is revised better than the old one.
Additional comments on the determination of sample size as described in the Methods section. Usually when determining sample size, population information and effect sizes based on previous studies related to the primary outcome are required. In this study, it states that it is determined only by alpha and beta values, which I believe is not possible.
Author Response
All my comments seem to be appropriately addressed and the manuscript is revised better than the old one.
1. Additional comments on the determination of sample size as described in the Methods section. Usually when determining sample size, population information and effect sizes based on previous studies related to the primary outcome are required. In this study, it states that it is determined only by alpha and beta values, which I believe is not possible.
Response 1. We fully accepted the reviewer's suggestion and explained in detail the method of determining the sample size in the Section Materials and Methods, subsection Study Flow. Namely, we provide anticipated main outcome measure, alpha error, test power (beta error of 0.1), and minimum estimated number of subjects in each study group.